# Long-Term Effect of Exercise on Irisin Blood Levels—Systematic Review and Meta-Analysis

**DOI:** 10.3390/healthcare9111438

**Published:** 2021-10-25

**Authors:** Tereza Jandova, Angel Buendía-Romero, Hana Polanska, Veronika Hola, Marcela Rihova, Tomas Vetrovsky, Javier Courel-Ibáñez, Michal Steffl

**Affiliations:** 1Faculty of Physical Education and Sport, Charles University, 16252 Prague, Czech Republic; tjandova@ftvs.cuni.cz (T.J.); hpolanska@ftvs.cuni.cz (H.P.); hola@ftvs.cuni.cz (V.H.); tvetrovsky@ftvs.cuni.cz (T.V.); 2Faculty of Sport Sciences, University of Murcia, 30720 Murcia, Spain; angel.buendiar@um.es (A.B.-R.); javier.courel.ibanez@gmail.com (J.C.-I.); 3Centre of Expertise Longevity and Long Term Care and Centre of Gerontology, Faculty of Humanities, Charles University, 18200 Prague, Czech Republic; m.molackova@email.cz

**Keywords:** physical activity, myokines, health, intervention

## Abstract

Physical exercise may activate a number of important biochemical processes in the human body. The aim of this systematic review and meta-analysis was to identify the long-term effect of physical activity on irisin blood levels. We searched PubMed, Scopus, and Web of Science for articles addressing the long-term effect of physical exercise on irisin blood levels. Fifty-nine articles were included in the final qualitative and quantitative syntheses. A statistically significant within-group effect of exercise on irisin blood levels was in 33 studies; out of them, the irisin level increased 23× and decreased 10×. The significant positive between-groups effect was found 11×. Furthermore, the meta-analysis indicated that physical exercise had a significant positive effect on irisin blood levels (SMD = 0.39 (95% CI 0.27–0.52)). Nevertheless, considerably high heterogeneity was found in all the analyses. This systematic review and meta-analysis indicate that physical exercise might increase irisin blood levels; however, the results of individual studies were considerably inconsistent, which questions the methodological detection of irisin by ELISA kits.

## 1. Background

### Description of the Condition

Irisin is a cleavage product of fibronectin type III domain-containing protein 5 (FNDC5) and was first isolated and described by Bostrom, Wu [1] as an exercise-induced hormone in 2012. Since then, many beneficial roles have been ascribed to it. For instance, as an important regulator of energy metabolism, irisin plays a protective role against type 2 diabetes mellitus and obesity [2,3]; maintains cardiovascular health [4]; acts as a behavioural antidepressant in mood regulation [5], and protects against bone loss and muscle atrophy [6]. Irisin has also been linked to the increased expression of brain-derived neurotrophic factor (BDNF) with subsequent beneficial effects on brain health and cognitive function [7]. Most recently, irisin has also shown a positive effect in regulating diverse genes in the adipose tissue, related to the COVID-19 outcome. Precisely, a reduction of genes implicated in elevated viral infection and an increase in genes that block virus-cell cleavage, which indicates a decrease of SARS-CoV-2 infection rate in human cells, has been demonstrated [8].

The irisin-mediated therapeutic effect may, in fact, hold the answers to how physical exercise positively influences the human body. Indeed, it is common knowledge that physical exercise helps to improve health status and can help prevent many diseases such as cardiovascular diseases [9], insulin resistance [10], type 2 diabetes mellitus [11], depression [12], sarcopenia [13] or Alzheimer’s disease [14]. Exercise stimulates PPARγ coactivator-1 α (PGC1α) as a transcriptional coactivator that mediates many biological programs related to energy metabolism. More specifically, it stimulates the expression of FNDC5, which encodes a type I membrane protein that is processed proteolytically, resulting in irisin secretion into the blood [1]. Therefore, any effect of FNDC5 and resulting irisin in regulating the health benefits of exercise is likely dependent upon its induction by exercise.

Nevertheless, contradictory findings have been emerging concerning the function of irisin, its precursor gene, and the relationship between PGC-1α and FNDC5 expression [15,16,17]. For example, Pekkala et al. [15] in 2013 (one year after the first isolation of irisin) found that the upregulation of PGC-1α mRNA expression did not correspond with the FNDC5 mRNA upregulation. Moreover, several experimental studies focused on both the acute and long-term effects of physical exercise on blood irisin levels in different contexts have been recently reviewed, still leaving us with inconclusive results [18]. Therefore, the main aim of this systematic review and meta-analysis was to investigate the long-term effect of physical exercise on blood irisin levels in order to identify a common effect, which could shed a better light on such phenomenon, especially when considering its role in health as a potential therapeutic target and further research in this area.

## 2. Materials and Methods

This systematic review and meta-analysis were conducted according to the recommendations and criteria outlined in the Preferred Reporting Items for Systematic Reviews and Meta-Analyses (PRISMA) [19].

### 2.1. Criteria for Considering Studies for this Review

Studies focused on the influence of physical activity on blood irisin concentration were considered in the analysis. Papers had to be written in English and published in peer-reviewed journals between 2012 and 2021.

### 2.2. Types of Studies

Randomized control trials (RCT), experimental or semi-experimental studies were considered for this study.

### 2.3. Types of Participants

All participants, including males and females, were considered for this study without regard to age or health conditions.

### 2.4. Types of Interventions

All physical activities such as endurance, resistance exercise, walking, dancing, etc., were considered for this study. There were no limitations concerning the duration of the intervention.

### 2.5. Types of Outcome Measures

The outcome measure was irisin in the bloodstream—a continual scale with physical activities as factors.

### 2.6. Primary Outcomes

Blood irisin level was measured using a standardized commercial Enzyme-Linked Immunosorbent Assay (ELISA) kit.

### 2.7. Search Methods for Identification of Studies 

Appropriate papers were identified through searches using two electronic databases: PubMed, Scopus, and a metasearch engine: Web of Science. Additionally, the reference lists of eligible papers and several recently published reviews were hand-searched for further studies. The search stream used in all the databases is presented in Table 1.

### 2.8. Data Collection and Analysis

All potential papers were first downloaded in EndNote, and then all duplicates were deleted. After removing all the duplicates, all abstracts were explored to identify relevant papers for subsequent selections. If from the abstract the papers seemed suitable, full texts were examined in detail. Additionally, other papers were identified through the reference lists of papers and reviews gained by the database search. 

### 2.9. Assessment of Risk of Bias in Included Studies

A modified version of the Cochrane risk of bias tool (RoB 2) for randomized [20], and risk of bias in non-randomized studies—of interventions (ROBINS-I) for non-randomized comparative studies was used to assess the methodological quality of the included studies [21].

### 2.10. Measures of Treatment Effect

We calculated the standardized mean difference for each study, and then the Cochran–Mantel–Haenszel statistical method based on a fixed-effect model was used to calculate an effect size [22]. We estimated the heterogeneity using the Cochran Q statistic and I^2^. A rough guide to the interpretation of I^2^ is as follows: 0 to 40% might not be important, 30% to 60% may represent moderate heterogeneity, 50% to 90% may represent substantial heterogeneity, and 75% to 100% represents considerable heterogeneity [23]. Statistics were carried out using Review Manager 5.4.

### 2.11. Dealing with Missing Data

To calculate the standardized mean difference, in our case Hedges’ *g*, we needed the sample size for the experimental and control group and the above-mentioned mean differences (after—before) with SD for both groups. In case that they were not available, we calculated them using baseline and follow-up means and SD as a simple post—pre difference; we estimated SD as [24]:$$SD_{E,\;change}=\sqrt{SD_{E,\;baseline}^{2}+SD_{E,final}^{2}-\left(2\times Corr\times SD_{E,baseline}\times SD_{E,final}\right)}$$

The correlation coefficient—Corr was calculated using this formula: $$Corr_{E}=\frac{SD_{E,baseline}^{2}+SD_{E,final}^{2}-SD_{E,change}^{2}}{2\times SD_{E,baseline}\times SD_{E,final}}$$

## 3. Results

### 3.1. Description of Studies 

Figure 1 summarises the yield of the search process. Fifty-nine studies were included in this systematic review involving 2164 participants. Healthy participants were included in 32 studies, and three studies were focused on elite or sub-elite athletes. Patients suffering from several different diseases (interstitial lung disease, progressive multiple sclerosis, or type II diabetes) participated in nine studies. Seventeen studies were focused on obese or overweight participants and one on pregnant females. The average age ranged between 9 and 71 years. The basic description of the included studies is presented in Table 2.

### 3.2. Risk of Bias and Quality of Reporting Data

Twenty-eight studies were randomized, and out of these, three did not use any non-exercised control group. The randomized studies showed a relatively low risk of bias according to RoB 2; nevertheless, no studies were without any risk of bias. Almost all of the included studies showed a low risk of bias of “selective reporting” because they reported all the outcomes measured, and all the included studies displayed a low risk of bias in “other bias” (Figure 2). Thirty-one studies were not randomized, and out of these, 14 were a single group design study. The metrological quality of the non-randomized studies was relatively high. The risk of bias assessment of the included papers using the ROBINS-I tool for non-randomized comparative studies is presented in Figure 3.

### 3.3. Systematic Review

Twenty-five studies were randomized control trials (RCT). The rest used different approaches, such as non-randomized trials, often without any non-exercised control group. Several different physical activities (endurance or resistance exercise, walking, swimming, etc.) were used in the studies. A 33× statistically significant within-group effect was found, where the irisin level increased 23× and decreased 10×. The statistically significant between-groups effect was found in 15 studies. A significant positive effect after exercise compared to non-exercise control groups was found in endurance training (ET) 4×, in resistance training (RT) 3×, and in combined training (CT), high-intensity interval training (HIIT), concurrent aerobic-resistance (A-R) as well as in concurrent resistance-aerobic (R-A) training 1×. A significant positive effect was also found after high-intensity interval training (HIIT) compared to continuous moderate-intensity training (CMIT). Long-term moderate physical exercise had a significant positive effect on the irisin blood level in obese compared to normal-weight adults as well as RT in older adults. On the other hand, taekwondo decreased the irisin level in obese children as well as high-repetition resistance training (HRRT) in healthy adults. A lower effect in the irisin level had RT compared to RT with ursolic acid (UA) supplementation in healthy adults. The results of these studies focused on the long-term effect are shown in Table 3.

### 3.4. Meta-Analysis

Data from 717 participants in the exercise groups and 467 participants in the non-exercise control groups were included in the overall effect of the meta-analysis. The overall effect was statistically significant, favoring the exercise group (SMD = 0.39 (95% CI 0.27–0.52)). We divided all the control studies into five groups according to the age and diseases presence for the other analyses. A statistically significant positive effect on irisin blood levels was found in healthy older adults (SMD = 0.32 (95% CI 0.11–0.54)), in the obese older adults (SMD = 0.91 (95% CI 0.69–1.13)), and in obese young adults (SMD = 0.67 (95% CI 0.02–1.32)). Nevertheless, there was no effect found in unhealthy older adults. In healthy young adults, the effect tended to be negative (SMD = −0.24 (95% CI −0.54–0.06)). Unfortunately, heterogeneity except in unhealthy older adults was considerably high in all the analyses. The forest plots are shown in Figure 4.

## 4. Discussion

### 4.1. Summary of Main Results and Interpretations

To our knowledge, this is the first systematic review and meta-analysis that focused on the long-term effect of different exercise interventions on blood irisin levels, which included many rigorous studies. The results of this present study indicate that: (a) physical exercise might increase irisin blood levels in specific populations, including healthy and obese older adults as well as obese young adults; however, (b) the results of individual studies exhibit high heterogeneity. More specifically, from the 59 articles included in the final qualitative and quantitative analyses, there was a statistically significant within-group effect of exercise on irisin blood levels in 33 studies (the irisin level increased in 23 and decreased in 10), and the significant positive between-groups effect was found 11 times. These contradictions in the research results of different studies are discussed below. 

Considering the results of the studies analyzed in this systematic review and meta-analysis, it seems that long-term physical exercise increases irisin blood levels, especially in obese individuals. More specifically, our study shows that endurance training and combined training increase irisin blood levels in obese older females [26], combined training in obese middle-aged males [33], endurance [46,68] and resistance exercise [49] in young obese adult males, and resistance exercise training in obese older males [52,76]. Moreover, exercise and dietary lifestyle programs demonstrated increases in irisin blood levels in obese children [31], and pelvic floor muscle training increased irisin in obese older females with stress urinary incontinence [80]. As mentioned in the introduction, irisin is secreted from muscles in response to exercise and is believed to positively affect many physiological processes in the human body, such as inhibition of adipogenesis in the adipose tissue [83]. Therefore, for the obese population, irisin could be seen as a possible therapeutic target that positively affects resting energy, glucose tolerance, and insulin sensitivity [84]. In fact, high-intensity interval training increased irisin blood levels in older adults with type II diabetes [40], and concurrent aerobic-resistance training in older adults males with type II diabetes [58]. However, some data also show negative correlations between elevated irisin and adiposity [3]; therefore, for now, the role of irisin in obese and diabetic patients remains unclear. Further investigations are needed to elucidate the complexity of irisin interactions with these metabolic endpoints before considering irisin as a therapeutic target in patients with obesity or diabetes mellitus. 

Several kinds of physical exercise increased irisin blood levels among healthy people as well. High-intensity interval training increased irisin in young, healthy males [41,60] and healthy young adults [43], military aerobic training [48] increased irisin blood levels in young, healthy males and one study with a single group design found a significant increase in irisin in young females after treadmill exercise [37]. On the other hand, cycle ergometer—sprint training, consecutive training [77] as well as indoor running [64] led to a decrease of irisin blood levels in young, healthy males, and the same effect was found after basketball training in healthy children [39] or after taekwondo training in obese children [70]. A decrease in irisin blood levels was also found after high-intensity interval training [56], CrossFit training [59], resistance training, and concurrent training [72] in healthy young females as well as after a structured group fitness program in pregnant females [75]. Climbing 4000 m peaks in the Mont Blanc massif led to a decrease of irisin blood levels as well [74]. Currently, no studies are reporting on the beneficial or possible negative effects of increased or decreased irisin levels in the young, healthy population, except for that in athletes, where the irisin level was positively correlated with bone strength [3]. However, we also postulate that irisin may exhibit prophylactic effects against metabolic disorders such as obesity or type II diabetes in this population.

More importantly, exercise including resistance training [44], aquaerobic training [50], endurance training [55], golf [61], low-intensity resistance training [66], high-intensity interval training [69], and treadmill walking [51] increased irisin blood levels in healthy older females and resistance training [82] in healthy older males, which may, in fact, protect against bone loss and muscle atrophy [6]. In this case, irisin might provide a therapeutic choice for treating diseases caused by inactivity (which often is the case for older adults), including osteoporosis and sarcopenia, or it may be used as a useful biomarker for the assessment of bone and muscle health as suggested by Leustean et al. [85]. Especially, the older adult female population would benefit from this protective effect as they are prone to osteoporosis. Cosio et al. [86] and Morteza et al. [87] also showed greater increases of circulating irisin in older adults after resistance training programs. It would also be worth investigating the effect of neuromuscular electrical stimulation (NMES; a passive type of exercise) protocols, such as in the study by Jandova et al. [88], on circulating irisin, which could offer an alternative mode of exercising for people with difficulties maintaining a physically active life. In terms of aerobic training, Morteza et al. [87] reported no significant changes for circulating irisin, which is contrary to our findings. Instead, we report on decreases in circulating irisin in blood following resistance training in normal weighted older males [76] and after high-repetition resistance training in healthy older adults [73]. In any case, the overall effect of the meta-analysis was statistically significant, favouring the exercise group (SMD = 0.39 (95% CI 0.27–0.52)), which indicates that the long-term effect of exercise on blood irisin levels is more positive regardless of the type of training. In that case, qualified physical exercise professionals could use different training strategies based on the needs and preferences of individuals. 

Two main caveats must be considered when interpreting the findings of this review and meta-analysis—the higher overall heterogeneity and the methodological aspects of measuring irisin in blood, which may be the actual culprit of the high heterogeneity. Currently, all evidence for irisin in the blood in this review is based on commercial ELISA kits. These kits are based on polyclonal antibodies (pAbs), which have been recently found to have prominent cross-reactivity with non-specific proteins in human and animal sera [89]. Such findings question all previous data obtained with these kits, and until these methodological aspects are resolved, studies relying on these measures should be carefully scrutinized. In summary, this review and meta-analysis may have some evidentiary support for the positive long-term effects of exercise on blood irisin levels, but more importantly, it points to the important methodological issues related to the actual detection of irisin [90], which should be considered when conducting research in this area [85]. In summary, our results confirm the general notion that exercise increases irisin levels in obese and older people. Additionally, irisin may possess protective properties against obesity and possibly also against osteoporosis or sarcopenia, which needs to be investigated further by future studies.

### 4.2. Quality of the Evidence

The studies included in this review and meta-analysis were comprised of both randomized and non-randomized trials (28 vs. 31, respectively) and were assessed as low risk for bias and quality of reporting data. All included studies assessed effects on the completers only, which may result in an overstatement of the effects.

### 4.3. Potential Biases in the Review Process

This systematic review and meta-analysis are limited to published research, and no unpublished studies were included in this review. Therefore, our review may be biased due to the possible threat of publication and reporting bias. 

### 4.4. Agreements and Disagreements with Other Studies or Reviews

The current findings are generally consistent with prior reviews [18,86,91,92,93] in that the effects of physical exercise on blood irisin levels vary. Although prior reviews describe the positive effects of both acute and chronic (long-term) effects of physical exercise, they also point to mixed results and the overall lack of evidence [18]. This work, however, represents a more rigorous extension of existing work as a meta-analysis was carried out, demonstrating that the overall effect of exercise on irisin blood levels was statistically significant.

## 5. Conclusions

In summary, the present systematic review and meta-analysis indicate that long-term physical exercise might increase irisin blood levels in the populations of healthy, obese older adults, and young obese adults. On the other hand, in the population of ill older adults, the results of this study indicate no effect and even a decrease in circulating irisin for healthy young people. These contradictory results and large heterogeneity found in this study, along with many other contradictions brought by previous research in this area, including the methods of irisin detection, call for further investigations. In addition, although many studies in this review and meta-analysis demonstrated a low risk of bias, future research in this area would benefit from more sophisticated and rigorous designs. Larger sample sizes would permit analyses that can account for the heterogeneity, which is an essential factor. Nevertheless, the main focus should be placed on improving current analytical techniques to measure blood irisin levels, which may be the main caveat in this research area before considering irisin as a therapeutic target.

## Figures and Tables

**Figure 1 healthcare-09-01438-f001:**
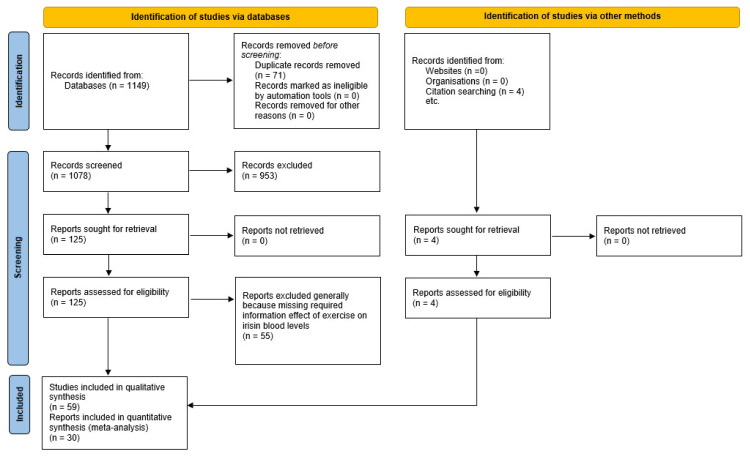
Flowchart illustrating the different phases of the search and study selection.

**Figure 2 healthcare-09-01438-f002:**
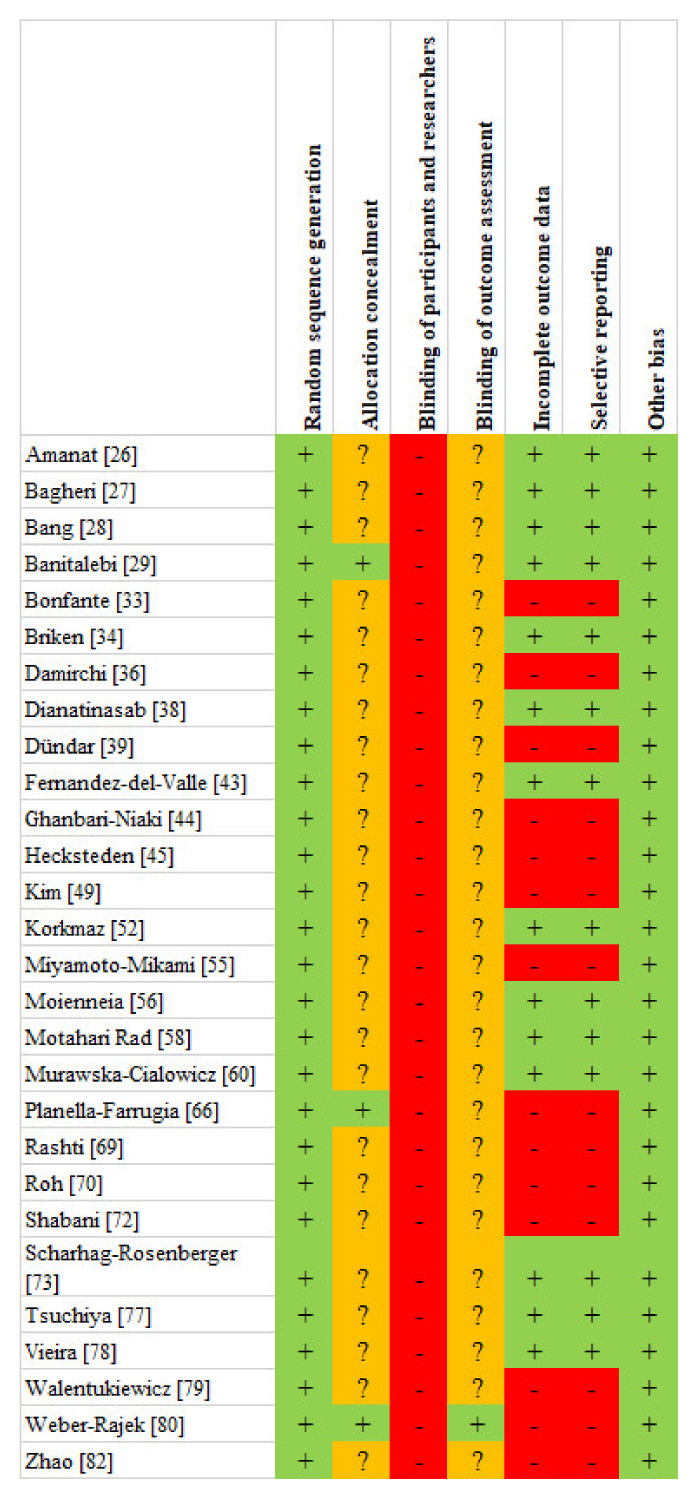
Risk of bias assessment using the RoB 2 for randomized control trials. *Note*: + Low risk; ? Some concerns; - High risk.

**Figure 3 healthcare-09-01438-f003:**
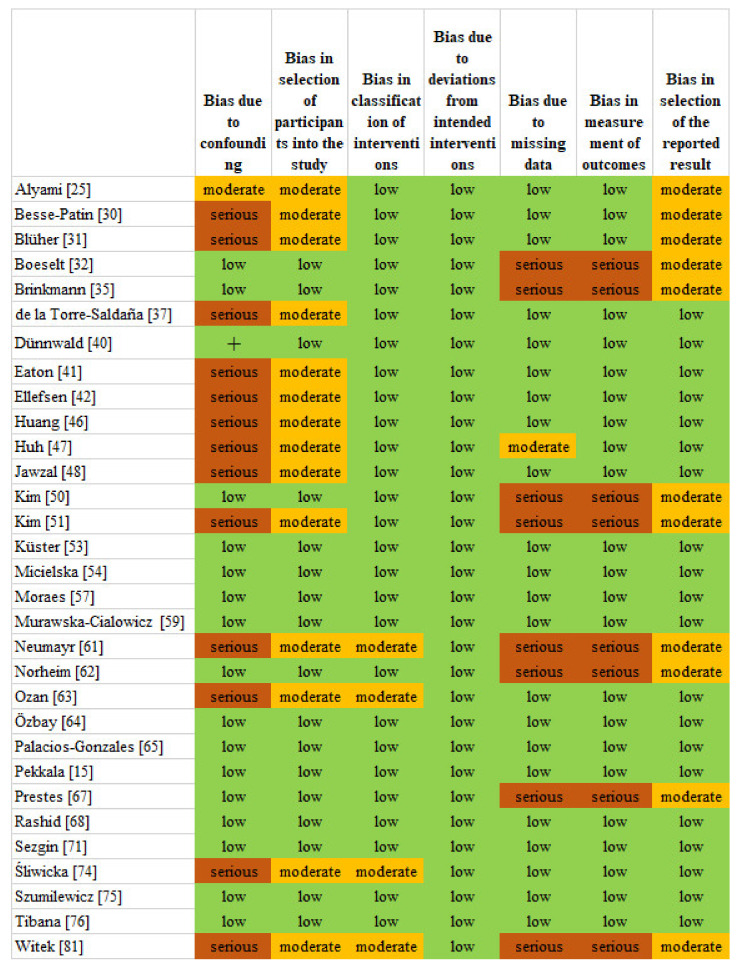
Risk of bias assessment using the ROBINS-I tool for non-randomized comparative studies.

**Figure 4 healthcare-09-01438-f004:**
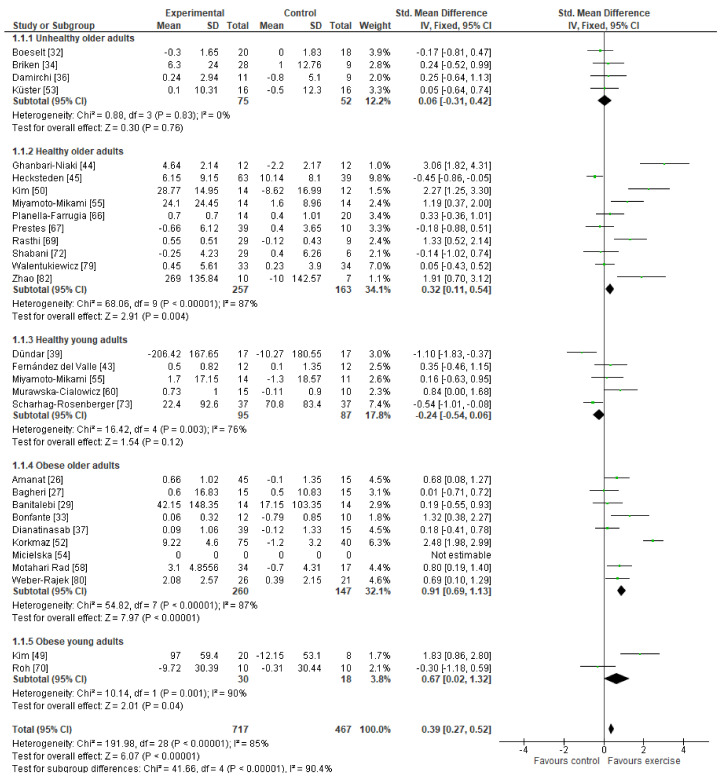
The forest plots of effect sizes for all the groups.

**Table 1 healthcare-09-01438-t001:** Search results from electronic databases.

PubMed Central	Search ((((irisin) OR FNDC5)) AND ((exercise) OR physical)) NOT (((((mice) OR rats) OR mouse) OR rodents) OR animal)	236
SCOPUS	((TITLE-ABS-KEY (irisin) OR TITLE-ABS-KEY (FNDC5))) AND ((TITLE-ABS-KEY (exercise) OR TITLE-ABS-KEY (physical))) AND NOT ((TITLE-ABS-KEY (mice) OR TITLE-ABS-KEY (rats) OR TITLE-ABS-KEY (mouse) OR TITLE-ABS-KEY (rodents) OR TITLE-ABS-KEY (animal)))	386
WoS	TOPIC: (irisin) OR TOPIC: (FNDC5) AND TOPIC: (exercise) OR TOPIC: (physical) NOT TOPIC: (mice) OR TOPIC: (rats) OR TOPIC: (mouse) OR TOPIC: (rodents) OR TOPIC: (animal)	527

**Table 2 healthcare-09-01438-t002:** Studies included in this systematic review.

Study	Year	Country	Design	Sample Description	Sample Size	Sex	Mean Age (SD) of the Whole Sample
Alyami [25]	2020	Saudi Arabia	NRT; N-CG	Interstitial lung disease	10	T	30–40+
Amanat [26]	2020	Iran	RCT	Obese with metabolic syndrome	60	F	54.5 (6.9)
Bagheri [27]	2020	Iran	RCT	Obese	30	M	43.8 (3.4) *
Bang [28]	2020	South Korea	RT; N-CG	Healthy	16	M	29.4 (5.1)
Banitalebi [29]	2019	Iran	RCT	Obese with type II diabetes	52	F	55.4 (5.9) *
Besse-Patin [30]	2014	France	SGS	Obese	11	M	35.4 (1.5)
Blüher [31]	2014	Germany	SGS	Obese	65	T	12.5 (1.6)
Boeselt [32]	2017	Germany	NRT	Chronic obstructive pulmonary disease	37	T	65.7 (8.3)
Bonfante [33]	2017	Brazil	RCT	Obese	22	M	49.1 (5.8)
Briken [34]	2016	Germany	RCT	Progressive multiple sclerosis	42	T	50.0 (7.5)
Brinkmann [35]	2020	Germany	NRT; N-CG	Obese	22	T	46–74
Damirchi [36]	2018	Iran	RCT	Mild cognitive impairment	20	F	68.8 (3.7) *
de la Torre-Saldaña [37]	2019	Mexico	SGS	Healthy	38	F	23.0 (3.3)
Dianatinasab [38]	2020	Iran	RCT	Obese with metabolic syndrome	54	F	53.5 (6.5)
Dündar [39]	2019	Turkey	RCT	Healthy	34	M	14.5 (1.1)
Dünnwald [40]	2019	Austria	NRT; N-CG	Type II diabetes	14	T	59.6 (5.7) *
Eaton [41]	2017	Canada	SGS	Healthy	9	M	20.5 (1.5)
Ellefsen [42]	2014	Germany	SGS	Healthy	18	F	26.0 (6.0)
Fernandez-del-Valle [43]	2018	US	RCT	Healthy	26	T	21.2 (1.9) *
Ghanbari-Niaki [44]	2018	Iran	RCT	Healthy	24	F	55.7 (4.9)
Hecksteden [45]	2013	Germany	RCT	Healthy	102	T	49.0 (7.0) *
Huang [46]	2017	China	SGS	Obese	22	T	22.1 (2.8)
Huh [47]	2014	US	SGS	Healthy	14	F	24.3 (2.6)
Jawzal [48]	2020	Kurdistan	SGS	Healthy	39	M	24 (22–27)
Kim [49]	2016	South Korea	RCT	Obese	28	T	25.7 (4.1) *
Kim [50]	2018	South Korea	NRT	Healthy	26	F	71.8 (3.1) *
Kim [51]	2020	South Korea	SGS	Healthy	25	F	60.3 (5.3)
Korkmaz [52]	2019	Finland	RCT	Obese	144	M	40–65
Küster [53]	2017	Germany	NRT	Mild cognitive impairment	46	T	71.2 (6.0)
Micielska [54]	2019	Poland	NRT	Healthy	33	F	40.0 (11.0) *
Miyamoto-Mikami [55]	2015	Japan	RCT	Healthy	53	T	21.0 (1.0)67.0 (8.0)
Moienneia [56]	2016	Iran	RCT	Healthy	21	F	24.4 (3.0)
Moraes [57]	2013	Brazil	NRT; N-CG	Hemodialysis patients	26	T	44.8 (14.1)
Motahari Rad [58]	2020	Iran	RCT	Type II diabetes	51	M	43.9 (2.5) *
Murawska-Cialowicz [59]	2015	Poland	SGS	Healthy	12	T	26.8 (6.8) *
Murawska-Cialowicz [60]	2020	Poland	RCT	Healthy	25	M	32.4 (6.6) *
Neumayr [61]	2020	Austria	SGS	Healthy	52	T	54.3
Norheim [62]	2013	Norway	NRT; N-CG	Prediabetes	26	M	40–65
Ozan [63]	2020	Turkey	SGS	Elite boxers	9	M	17.2 (3.3)
Özbay [64]	2020	Turkey	NRT; N-CG	Healthy	33	M	22.6 (1.6) *
Palacios-Gonzales [65]	2015	Mexico	NRT; N-CG	Obese	85	T	9.0 (0.9) *
Pekkala [15]	2013	Finland	NRT; N-CG	Healthy	63	M	24–68
Planella-Farrugia [66]	2019	Spain	RCT	Healthy	43	T	71.2 (3.3) *
Prestes [67]	2015	Brazil	NRT	Healthy	59	F	69.2 (6.1) *
Rashid [68]	2020	Iraq	NRT; N-CG	Obese	60	M	20–43
Rashti [69]	2019	Iran	RCT	Healthy	48	F	57.1 (4.1) *
Roh [70]	2020	South Korea	RCT	Obese	20	T	12.6 (0.5)
Sezgin [71]	2020	Turkey	NRT; N-CG	Obese	37	F	47.9 (13.2)
Shabani [72]	2018	Iran	RCT	Healthy	31	F	24.6 (2.5) *
Scharhag-Rosenberger [73]	2014	Germany	RCT	Healthy	74	T	47.0 (7,0)
Śliwicka [74]	2017	Poland	SGS	Climbers	8	M	27.0 (2.8)
Szumilewicz [75]	2017	Poland	NRT; N-CG	Pregnant	9	F	23.0 (3.0)
Tibana [76]	2017	Brazil	NRT; N-CG	Obese	49	F	61–68
Tsuchiya [77]	2016	Japan	RT; N-CG	Healthy	20	M	20.4 (0.8) *
Vieira [78]	2020	Brazil	RT; N-CG	Healthy	20	F	64.1 (7.0) *
Walentukiewicz [79]	2018	Poland	RCT	Healthy	94	F	68.0 (5.1)
Weber-Rajek [80]	2019	Poland	RCT	Obese with stress urinary incontinence	49	F	62.5 (2.0) *
Witek [81]	2016	Poland	SGS	Tennis players	12	M	16.0 (2.0)
Zhao [82]	2017	China	RCT	Healthy	17	M	62.3 (3.5) *

*Note*: * exercise group; SD = standard deviation; NRT = nonrandomised trial; N-CG = no control group; RCT = randomised controlled trial; RT = randomised trial; SGS = single group design study.

**Table 3 healthcare-09-01438-t003:** Studies focused on the long-term effect of physical exercise on blood irisin levels.

Study	Intervention Description	Length of the Intervention	Weekly Volume	Within-Group Effect Sig.	Between-Groups Effect Sig.	Note
Alyami [25]	Supervised exercise training (SET)	8 weeks	2×	-	-	
Amanat [26]	Endurance training (ET), resistance training (RT), and combined training (CT)	12 weeks	2× to 3×	↑ *	↑ **	* all the EG; ** ET and CT
Bagheri [27]	Endurance training (ET)	8 weeks	3×	-	-	
Bang [28]	Resistance training (RT) vs. resistance training with ursolic acid supplementation (RT + UA)	8 weeks	6×	-	↓ *	* RT
Banitalebi [29]	Sprint interval training (SIT), combined endurance and resistance training (A + R)	10 weeks	3×	-	-	
Besse-Patin [30]	Endurance training	8 weeks	5×	-	N/A	
Blüher [31]	Exercise and dietary lifestyle program	1 year	2×	↑	N/A	
Boeselt [32]	High-intensity training (HIT)	12 weeks	2×	-	-	
Bonfante [33]	Combined training (CT)	24 weeks	3×	-	↑	
Briken [34]	Endurance training (ET)	9 weeks	2–3×	-	-	
Brinkmann [35]	Combined training: males vs. females	8 weeks	3×	-	-	
Damirchi [36]	Physical training (PT)	8 weeks	2×	-	-	
de la Torre-Saldaña [37]	Treadmill—6.0–7.9 METs and >8.0 METs	2 weeks	5×	↑ *	N/A	* both
Dianatinasab [38]	Endurance training (ET), resistance training (RT), and combined training (CT)	8 weeks	3×	-	-	
Dündar [39]	Basketball training	8 weeks	5×	↓	-	
Dünnwald [40]	High-intensity interval training (HIIT) vs. continuous moderate-intensity training (CMIT)	4 weeks	3×	↑ *	↑ *	* HIIT
Eaton [41]	High-intensity interval training (HIIT)	20 days	2× a day	↑	N/A	
Ellefsen [42]	Progressive strength training	12 weeks	3×	-	N/A	
Fernandez-del-Valle [43]	High-intensity interval training (HIIT)	3 weeks	3×	↑	↑	
Ghanbari-Niaki [44]	Resistance training (RT)	9 weeks	3×	↑	-	
Hecksteden [45]	Endurance training (ET) and strength training (ST)	26 weeks	3×	-	-	
Huang [46]	Endurance exercise	8 weeks	7×	↑	N/A	
Huh [47]	Whole-body vibration exercise	6 weeks	2×	-	N/A	
Jawzal [48]	Military aerobic training	8 weeks	7×	↑	N/A	
Kim [49]	Endurance training (ET), resistance training (RT)	8 weeks	5×	↑ *	↑ *	* RT
Kim [50]	Aquaerobic training (AqT)	16 weeks	2×	↑	↑	
Kim [51]	Treadmill walking	6 weeks	3×	↑	N/A	
Korkmaz [52]	Nordic walking (NW), resistance exercise (RE)	12 weeks	3×	↑ *	↑ *	* Both IG
Kuster [53]	Physical training (PT)	10 weeks	2×	-	-	
Micielska [54]	High-intensity circuit training (HICT)	5 weeks	4×	-	-	
Miyamoto-Mikami [55]	Endurance training (ET)—healthy young	8 weeks	3×	-	-	
	Endurance training (ET)—middle-aged/older	8 weeks	3×	↑	↑	
Moienneia [56]	Resistance training low (LIRT) vs. high intensity (HIRT)	8 weeks	3×	↓ *	-	* HIRT
Moraes [57]	Intradialytic resistance training (IRT)	6 months	3×	↑	N/A	
Motahari Rad [58]	Concurrent aerobic-resistance (A-R) and concurrent resistance-aerobic (R-A) training	12 weeks	3×	↑ *	↑ *	* Both IG
Murawska-Cialowicz [59]	CrossFit training: males vs. females	3 months	2×	↓ *	-	* females
Murawska-Cialowicz [60]	High-intensity interval training (HIIT)	8 weeks	2×	↑ *	-	* HIIT
Neumayr [61]	Golf vs. Nordic walking or e-biking	1 week	7×	↑ *	N/A	* only golf group
Norheim [62]	Combined endurance and strength training: normoglycaemic and normal weight	12 weeks	4×	-	-	
Ozan [63]	Strength training with thera-band	8 weeks	3×	-	N/A	
Özbay [64]	Outdoor running (OR) vs. indoor running (IR)	18 weeks	4×	↓ *	-	* IR
Palacios-Gonzales [65]	School-based physical activity program: normal weight	8 months	5×	-	-	
Pekkala [15]	Endurance training (ET) vs. combined endurance and resistance training (ET + RT)	21 weeks	2× (ET) or 2× (ET) + 2× (RT)	-	-	
Planella-Farrugia [66]	Low-intensity resistance training (LIRT)	16 weeks	2×	↑ *	-	* LIRT
Prestes [67]	Resistance training linear periodization (LP) and undulating periodization (UP)	16 weeks	2×	-	-	
Rashid [68]	Long-term moderate physical exercise: normal weight	6 months	7 times	↑ *	↑	* both
Rashti [69]	High-intensity interval (HIIT) and moderate-intensity training (MIIT)	10 weeks	3×	↑*	-	* HIIT
Roh [70]	Taekwondo in obese children	16 weeks	5×	↓	↓	
Sezgin [71]	Endurance training (ET) and personalized nutrition programs: normal weight	8 weeks	7×	-	-	
Shabani [72]	Resistance training (RT), Endurance training (ET), and concurrent (endurance + resistance) training (CT)	8 weeks	3×	↓ *	-	* RT and CT
Scharhag-Rosenberger [73]	High-repetition resistance training (HRRT)	6 months	3×	-	↓	
Śliwicka [74]	Climb 4000 m peaks in the Mont Blanc massif	14 days	7×	↓	N/A	
Szumilewicz [75]	Structured group fitness program—very active (VA) vs. less active (LA) groups	8 weeks	≥3× (VA) <3× (LA)	↓ *	-	* LA
Tibana [76]	Resistance training (RT): obese vs. normal weight	16 weeks	3×	↓ *	↑ **	* normal weight; ** obese
Tsuchiya [77]	Cycle ergometer—sprint training (ST) vs. two consecutive training (TCT)	4 weeks	5× (ST) 2–3× (TCT)	↓ *	-	* both
Vieira [78]	Resistance training very high supervision (VHS) vs. high supervision (HS)	16 weeks	2×	-	-	
Walentukiewicz [79]	Nordic walking (NW)	12 weeks	3×	-	-	
Weber-Rajek [80]	Pelvic floor muscle training (PMT)	4 weeks	3×	↑ *	-	* PMT
Witek [81]	Workload during the competitive season	8 months	-	-	N/A	
Zhao [82]	Resistance training (RT)	12 weeks	2×	↑ *	↑	* RT

*Note*: ↑ = increased levels; ↓ = decreased levels; N/A not applicable; *, ** = groups’ specification.

## Data Availability

Data available in a publicly accessible repository that does not issue DOIs; Publicly available datasets were analyzed in this study. This data can be found here: https://uloz.to/tamhle/8qgosLnZa8BM/name/Nahrano-25-10-2021-v-12-01-24#!ZGVjBGR2A2V0ZQExLmtjLwOxAGNkA01PBTSFFaO0n2qjpwEwAN== (accessed on 22 June 2021).

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
