# Peer review of "Long-Term Effect of Exercise on Irisin Blood Levels—Systematic Review and Meta-Analysis"

_healthcare, 2021, doi:10.3390/healthcare9111438_

Round 1

Reviewer 1 Report

Dear authors, your manuscript has potential but there are important deficiencies that you must fix and/or justify before considering it for publication:

INTRODUCTION

  • The introduction is very short. Authors should include, for example, epidemiological data or previous research justifying the manuscript. It is not known why this manuscript is to be made.
  • The research objective is poorly stated. The objective cannot be to "summarize". The verb must be a concrete action in relation to IRISINA and what you want to investigate about it.

MATERIALS AND METHODS
Criteria for considering studies for this review:

  • The authors have not specified the chronological criteria. It is not known in what range of years the articles of its "results" have been searched.

Search methods for identification of studies:

  • WOS is not a database but a meta-search engine.

RESULTS

  • According to the flow chart, the authors have searched for articles in other ways but have not been described in the “methodology”. They must justify this.

DISCUSSION

  • The authors have not done the "discussion", but have used the references of their "results" to do the "discussion". This is wrong, it is redundant information. Authors should use other new references to justify their "results" without resorting to them.
  • The authors have not explained the implications for clinical practice of their data.

REFERENCES

  • There are errors in the references. Example, appointment 30 or 64 is incomplete. Authors should review the entire section.

Author Response

Dear reviewer, thank you for your valuable comments. It is very much appreciated it. We have addressed the issues accordingly (see individual comments below).

INTRODUCTION

  • The introduction is very short. Authors should include, for example, epidemiological data or previous research justifying the manuscript. It is not known why this manuscript is to be made.

The introduction was extended, including important previous research that discuss the controversy of the physiological role of FNDC5/irisin in mediating responses to exercise and the methods of irisin detection. Based on this, the aim was formulated more clearly.

  • The research objective is poorly stated. The objective cannot be to "summarize". The verb must be a concrete action in relation to IRISINA and what you want to investigate about it.

It has been addressed accordingly. With the help of meta-analysis, we wanted to investigate whether there is ‘a point’ to conduct further research in this area given the controversy and inconsistency behind such research.

MATERIALS AND METHODS
Criteria for considering studies for this review:

  • The authors have not specified the chronological criteria. It is not known in what range of years the articles of its "results" have been searched.

We have added this.

Search methods for identification of studies:

  • WOS is not a database but a meta-search engine.

Thank you for this comment. We have renamed it.

RESULTS

  • According to the flow chart, the authors have searched for articles in other ways but have not been described in the “methodology”. They must justify this.

We have added the information about it.

DISCUSSION

  • The authors have not done the "discussion", but have used the references of their "results" to do the "discussion". This is wrong, it is redundant information. Authors should use other new references to justify their "results" without resorting to them.

We tried to address this point by resorting to other reviews and meta-analyses in this area. However, for the summary purpose – we still left some of the references of our results in the discussion to detail the inconsistency of data.

  • The authors have not explained the implications for clinical practice of their data.

It has been addressed accordingly, pinpointing the implications for clinical practice based on our results for the population of obese and older individuals respectively.

REFERENCES

  • There are errors in the references. Example, appointment 30 or 64 is incomplete. Authors should review the entire section.

Thanks for this point, there were several errors in references. We have corrected them.

Reviewer 2 Report

1. Maybe „chronic effects“ is an inappropriate wording choice. Maybe the
authors decide to better speak about long-term effects?

2. The authors conclude that the high heterogeneity across analyses did
not allow drawing concrete conclusions. Maybe the relatively large
number of studies included may allow to run some moderator analyses, or
at least, to better group studies according to conceptual or
methodological factors. This may help to improve the impact of the paper.

3. The practical implications of a potential effect of exercise on
irisin blood levels could be better illustrated with more detailed,
specific examples.

4. Maybe the paper could elaborate the different conditions and
prerequisites of physical exercise in more detail. What about frail
adults that have difficulties to maintain a physically active life?

Author Response

Dear reviewer, thank you for your valuable comments. It is very much appreciated it. We have addressed the issues accordingly (see individual comments below).

  1. Maybe „chronic effects“ is an inappropriate wording choice. Maybe the authors decide to better speak about long-term effects?

It was changed to long-term, which actually sounds better; however, the term chornic is also widely used in the literature.

  1. The authors conclude that the high heterogeneity across analyses did not allow drawing concrete conclusions. Maybe the relatively large number of studies included may allow to run some moderator analyses, or at least, to better group studies according to conceptual or methodological factors. This may help to improve the impact of the paper.

That is a good point. Initially, we wanted to do the sensitivity analysis. However, the quality of RCT were similar. We also tried to divide studies according to several approaches that were used. Unfortunately, any manipulations or sensitivity analysis did not help in the terms of heterogeneity. Therefore, we finally decided not to include them. It seems that the problem really lies in the high heterogeneity that may be truthful (therefore, it is a result) and reflects perhaps the controversy of using ELISA as a method of detection of irisin (as mentioned in the discussion). Therefore, the meta-analysis was used to identify the reason for the variation.

  1. The practical implications of a potential effect of exercise on irisin blood levels could be better illustrated with more detailed, specific examples.

It has been addressed accordingly, pinpointing the implications for clinical practice based on our results for the population of obese and older individuals respectively.

  1. Maybe the paper could elaborate the different conditions and prerequisites of physical exercise in more detail. What about frail adults that have difficulties to maintain a physically active life?

Thank you for your valuable comment, we took it into consideration and elaborated upon it in the discussion. We suggested for example to investigate other modes of passive exercise such as NMES, which already showed positive results in the older population.

Round 2

Reviewer 1 Report

Dear authors,

you improved your manuscript enough, therefore it can be proposed for publication.